# The Effects of Hospitalisation on the Serum Metabolome in COVID-19 Patients

**DOI:** 10.3390/metabo13080951

**Published:** 2023-08-16

**Authors:** Tim Hensen, Daniel Fässler, Liam O’Mahony, Werner C. Albrich, Beatrice Barda, Christian Garzoni, Gian-Reto Kleger, Urs Pietsch, Noémie Suh, Johannes Hertel, Ines Thiele

**Affiliations:** 1School of Medicine, University of Galway, H91 TK33 Galway, Ireland; t.hensen1@universityofgalway.ie; 2School of Microbiology, University of Galway, H91 TK33 Galway, Ireland; 3Ryan Institute, University of Galway, H91 TK33 Galway, Ireland; 4APC Microbiome Ireland, T12 K8AF Cork, Ireland; liam.omahony@ucc.ie (L.O.); werner.albrich@kssg.ch (W.C.A.); 5Department of Psychiatry and Psychotherapy, University Medicine Greifswald, 17475 Greifswald, Germany; daniel.faessler@uni-greifswald.de; 6Department of Medicine and School of Microbiology, University College Cork, T12 K8AF Cork, Ireland; 7Division of Infectious Diseases and Hospital Epidemiology, Cantonal Hospital St. Gallen, 9007 St. Gallen, Switzerland; 8Fondazione Epatocentro Ticino, Via Soldino 5, 6900 Lugano, Switzerland; beatrice.barda@hin.ch (B.B.); christian.garzoni@moncucco.ch (C.G.); 9Clinic of Internal Medicine and Infectious Diseases, Clinica Luganese Moncucco, 6900 Lugano, Switzerland; 10Division of Intensive Care, Cantonal Hospital St. Gallen, Rorschacherstrasse 95, 9007 St. Gallen, Switzerland; gian-reto.kleger@kssg.ch; 11Department of Anesthesia, Intensive Care, Emergency and Pain Medicine, Cantonal Hospital St. Gallen, Rorschacherstrasse 95, 9007 St. Gallen, Switzerland; urs.pietsch@kssg.ch; 12Division of Intensive Care, Geneva University Hospitals, The Faculty of Medicine, University of Geneva, 1211 Geneva, Switzerland; noemie.suh@hcuge.ch; 13DZHK (German Centre for Cardiovascular Research), Partner Site Greifswald, University Medicine Greifswald, 17475 Greifswald, Germany

**Keywords:** COVID-19, hospitalisation, metabolomics, serum, disease progression, multi-centre

## Abstract

COVID-19, a systemic multi-organ disease resulting from infection with severe acute respiratory syndrome coronavirus 2 (SARS-CoV-2), is known to result in a wide array of disease outcomes, ranging from asymptomatic to fatal. Despite persistent progress, there is a continued need for more accurate determinants of disease outcomes, including post-acute symptoms after COVID-19. In this study, we characterised the serum metabolomic changes due to hospitalisation and COVID-19 disease progression by mapping the serum metabolomic trajectories of 71 newly hospitalised moderate and severe patients in their first week after hospitalisation. These 71 patients were spread out over three hospitals in Switzerland, enabling us to meta-analyse the metabolomic trajectories and filter consistently changing metabolites. Additionally, we investigated differential metabolite–metabolite trajectories between fatal, severe, and moderate disease outcomes to find prognostic markers of disease severity. We found drastic changes in serum metabolite concentrations for 448 out of the 901 metabolites. These results included markers of hospitalisation, such as environmental exposures, dietary changes, and altered drug administration, but also possible markers of physiological functioning, including carboxyethyl-GABA and fibrinopeptides, which might be prognostic for worsening lung injury. Possible markers of disease progression included altered urea cycle metabolites and metabolites of the tricarboxylic acid (TCA) cycle, indicating a SARS-CoV-2-induced reprogramming of the host metabolism. Glycerophosphorylcholine was identified as a potential marker of disease severity. Taken together, this study describes the metabolome-wide changes due to hospitalisation and COVID-19 disease progression. Moreover, we propose a wide range of novel potential biomarkers for monitoring COVID-19 disease course, both dependent and independent of the severity.

## 1. Introduction

COVID-19 is a respiratory disease caused by infection with severe acute respiratory syndrome coronavirus 2 (SARS-CoV-2) [1]. Although most COVID-19 cases are mild, severe cases can result in acute lung injury [2] and acute respiratory distress syndrome [3], leading to potentially fatal outcomes [4]. COVID-19 has been reported to affect multiple organs beyond the lungs, including the gastrointestinal tract [5], the kidney [6], the liver [5,7], and the brain [8]. Due to increasing evidence on the systemic extrapulmonary implications of COVID-19 [9], the disease has been recognised as a multisystem disease [9]. Since the start of the COVID-19 pandemic, significant efforts have been made to find determinants of COVID-19 disease outcomes. Phenotypic and epidemiological determinants of COVID-19 disease outcomes, such as age [10,11], BMI [11,12,13], sex [14,15], air quality [16,17], and environmental pollution [14,15], have been instrumental in understanding which individuals are at risk of severe COVID-19 outcomes. These epidemiological metrics, however, are limited as they do not take genetics and molecular phenotypes into account. To find more accurate and time-dependent predictors of disease outcomes, omics-based studies, in particular blood metabolomics, have consistently found new prognostic markers of disease outcomes and enabled important insights into the pathogenesis of COVID-19 [18,19,20].

Previous blood metabolomic studies have uncovered widespread physiological and molecular responses upon infection by SARS-CoV-2 [21,22,23,24]. After recognition of the SARS-CoV-2 double-stranded RNA by the host, a series of molecular cascades is initiated, resulting in the production of cytokines and chemokines [1]. In severe and critical COVID-19 patients, the production of cytokines can become excessive, leading to a “cytokine storm”, which has been proposed as the main driver of COVID-19 severity [25]. IL-6, one of the main cytokines driving COVID-induced hyperinflammation [25], has been reported to correlate with shifts in tryptophan metabolism, nitrogen metabolism, and oxidative stress markers, such as methionine sulfoxide and cysteine. Notably, the depletion of tryptophan, an essential precursor of several neuroactive metabolites, such as serotonin and melatonin [26], has also been linked with persistent long-COVID symptoms.

Besides physiological and molecular changes due to the host response, SARS-CoV-2 has also been found to hijack the host metabolism to promote its replication [22,27,28]. SARS-CoV-2 is known to promote nucleotide production for its replication by increasing glucose-derived carbon uptake, leading to an increased oxidative tricarboxylic acid (TCA) cycle activity and nucleotide production [27]. Alterations in lipid metabolism have also been linked to SARS-CoV-2-induced metabolic hijacking [29,30,31]. SARS-CoV-2 replication is dependent on lipid droplets [32], which consist of various phospholipids, sphingolipids, and cholesterol compounds. Alterations in lipid metabolism have been confirmed by various metabolomic studies linking phospholipids and sphingolipids to COVID-19 disease outcomes [22,33,34]. 

COVID-19, being a systemic disease, impacts metabolism through COVID-19-related changes in the gut microbiome, with pathogenic gut microbes increasing in their abundance while beneficial microbes decrease in relative abundance [35]. Moreover, several gut-microbiome-modulated metabolites in the blood have been linked with COVID-19 outcomes, including short-chain fatty acids and neurotransmitters [35,36]. 

Although great strides have already been made to find determinants of COVID-19 disease outcomes, the interaction between hospitalisation and disease progression remains understudied. When an individual moves from the home environment to the hospital environment, the patient is often subject to a drastically different exposome, an umbrella term of all dietary, drug, behavioural, and environmental exposures [37]. The exposome shift from the home environment to the hospital can be expected to have systemic effects and may, thus, influence COVID-19 disease progression. 

To investigate how the hospital exposome, disease progression, and exposome–disease interactions influence physiological functioning, we analysed untargeted serum metabolomics data of newly hospitalised moderate and severe COVID-19 patients. Using a repeated-measure design, we investigated metabolome-wide shifts in the first week after hospitalisation of 71 patients in three different locations in Switzerland, namely Geneva, St. Gallen, and Ticino. Each location was investigated as a separate study, after which the results were meta-analysed to find consistently changing metabolites. We found drastic metabolome-wide changes in metabolism that could be linked to various aspects of hospitalisation and COVID-19 disease progression. This list included markers of changing environmental exposures, diet, drug metabolism, host–gut microbiota crosstalk, physiological functioning, and COVID-19-induced metabolic reprogramming. Moreover, we propose carboxyethyl-GABA and fibrinopeptides as potential markers of COVID-19-driven lung injury. This study comprehensively describes the shifts in serum metabolite concentrations in newly hospitalised COVID-19 patients and put these changes into their metabolic context. 

## 2. Results

### 2.1. Data Descriptions

To investigate serum metabolome trajectories during COVID-19, we analysed a total of 71 patients from three independent, longitudinal cohorts of COVID-19 patients placed in Ticino (*n* = 20), St. Gallen (*n* = 22), and Geneva (*n* = 29). We analysed serum samples using untargeted mass spectrometry during the first eight days of hospitalisation at two different time points. The Geneva and St. Gallen samples only consisted of severe COVID-19 cases, while the cohort in Ticino consisted of nine severe and eleven moderate COVID-19 cases (Table 1). Moderate cases were defined as PCR-confirmed SARS-CoV-2-infected patients with symptoms of pneumonia, fever, and respiratory tract symptoms. Severe COVID-19 patients had all the symptoms of moderate cases, but also had a respiratory rate of ≥30 breaths per minute and an oxygen saturation of ≤93% when breathing ambient air or having a PaO_2_/FiO_2_ below 300 mmHg. Patients that did not meet these requirements but needed ventilator support were also classified as severe COVID-19 patients. All moderate patients were situated in the hospital ward, whereas the severe patients were all in the intensive care unit (ICU). Across the three locations, 49 (37%) patients had one sample taken, 55 (42%) patients had two samples taken, 17 patients (13%) had three samples taken, and only nine patients (6%) had a fourth sample taken (Figure 1A).

The serum metabolome datasets contained measurements of 1086 different metabolites, consisting of a wide range of compounds with varied origins. These compounds included dietary markers, markers of environmental exposures, microbiome-derived metabolites, as well as a wide range of endogenously produced metabolites. Before analysing the metabolome datasets, we excluded all metabolites that were absent in over 20% of the samples in all three locations, resulting in 901 metabolites included in this study (Appendix A). The included metabolites represented nine global biochemical families, of which lipids (40% of all compounds) and amino acids and amino acid derivatives (20% of all compounds) were most numerous in terms of metabolic species (Figure 1B, Appendix A). The removed set of metabolites consisted mostly of lipids (41%) and unnamed metabolites (32%, Appendix A). Tryptophan was excluded from analysis, since its plasma levels dropped below the limit of detections in 67% of COVID-19 patients at the day of hospital admission, highlighting tryptophan depletion during COVID-19 as reported earlier [24].

### 2.2. The Serum Metabolome Drastically Changed within One Week Hospitalisation with COVID-19

We first analysed changes in the serum metabolome during the first eight days in the hospital. Therefore, we performed mixed-effect linear regression analysis for each sample and each metabolite, with the log-transformed concentration as the response variable and the time point as the predictor of interest (days after hospitalisation). Age, sex, BMI, and death due to COVID-19 were included as covariates with random intercepts for the individual. Next, we meta-analysed the regression outcomes to derive a list of consistently changing metabolites, reducing possible bias due to hospital-specific environments and treatments. After correction for multiple testing using the false discovery rate (FDR), 545 metabolites were significantly changed in the first eight days. After filtering out metabolites with heterogeneous effects (qFDR > 0.05) across the three samples (Appendix A), 448 metabolites consistently changed in their serum concentration. Each major biochemical class in the full metabolome data was included in these results, with lipids (163 compounds), amino acids (96 compounds), and xenobiotics (56 compounds) as the largest groups. Likewise, each pathway in the list of analysed metabolites was represented in the list of significant metabolites (Figure 1E), indicating large and widespread changes in physiological and metabolic functioning. The significantly changed metabolites point towards a range of physiological and metabolic processes previously linked with COVID-19 severity, but are also indicators of host–microbiome interactions, dietary changes, drug administration, and environmental exposures. Some of our top hits included carboxyethyl-GABA (increased), pantoate (increased), phenylacetylcarnitine (increased), and sphingomyelin (d18:1/25:0, d19:0/24:1, d20:1/23:0, d19:1/24:0) (increased, Figure 2A,B). Importantly, our results replicated with comparable effect sizes across the three different cohorts, indicating that these changes were independent of hospital-specific exposures. We further quantified our results by performing a pathway enrichment analysis on the 448 significantly and consistently changed metabolites using the MetaboAnalyst [38] pathway enrichment web service. Out of the 448 significant metabolites, 289 metabolites could be mapped onto the MetaboAnalyst 5.0 database (Appendix A). Pathway enrichment analysis then was performed against the KEGG homo sapiens reference pathway library [39], which resulted in eight enriched pathways after correcting for the false discovery rate. The KEGG arginine biosynthesis pathway, which included the urea cycle, was the top hit (eight hits, FDR = 1.11 × 10^−5^). The other enriched pathways were, in order of significance, aminoacyl-tRNA biosynthesis (ten hits, FDR = 0.006), panthothenate and CoA biosynthesis (six hits, FDR = 0.009), phenylalanine, tyrosine and tryptophan biosynthesis (three hits, FDR = 0.013), histidine metabolism (five hits, FDR = 0.019), caffeine metabolism (four hits, FDR = 0.019), beta-alanine metabolism (five hits, FDR = 0.0463), and sphingolipid metabolism (five hits, FDR = 0.046) (Appendix A).

In the following paragraphs, we will further highlight and contextualise the 448 measured serum biochemical changes. Please note that due to the large number of results, it was not possible to highlight and contextualise all results. Nevertheless, we tried to give a varied and broad overview of the measured biochemical changes with special attention to the effects of hospitalisation and COVID-19 disease progression (Appendix A). 

### 2.3. Serum Metabolome Trajectories Reflect Changing Environmental Exposures after Hospitalisation

Our results contained several markers of changing environmental exposures (Figure 3, Appendix A). For example, one marker was perfluorooctanoate (PFOA, decreased), which is an industrial surfactant and forever chemical used in the textile industry as a water and oil repellent coating [40]. Other markers of environmental exposure included propyl-4-hydroxybenzoate sulphate (decreased) and methyl-4-hydroxybenzoate sulphate (increased). Propyl-4-hydroxybenzoate and methyl-4-hydroxybenzoate are known as propylparaben and methylparaben, respectively, and both are widely used in cosmetics and body care [41], indicating changes in products used for body care in the hospital environment in comparison to the items utilised outside the hospital. 2,4-di-tert-butylphenol, an antioxidant with wide applications in industry [42], was also increased in the hospital setting. While the clinical meaning of those results is not clear, they suggest that changed environmental exposures due to hospitalisation affect the serum metabolome. 

### 2.4. Acetaminophen Metabolism Favoured Degradation to Glucuronide Conjugates in Lieu of Sulphate Conjugates

The metabolomics analyses also revealed changes in drug metabolites (Figure 4, Appendix A). Changes were found in both aspirin and paracetamol metabolism; both are wide-spread analgesics used in hospital and home settings. Salicylate, a downstream metabolite of acetyl-salicylic acid (aspirin), increased along with glucuronide conjugates of acetaminophen (paracetamol). The sulphate conjugates of acetaminophen contrastingly decreased in their concentrations. In conclusion, untargeted metabolomics identified metabolites of relevant drugs, indicating that the corresponding degradation pathways may be influenced by either the disease or changes in the environment due to hospitalisation.

### 2.5. Dietary Metabolites Indicate Changes in Diet in Hospitalised COVID-19 Cases

The changed serum metabolome trajectories also reflected changes in diet due to hospitalisation (Figure 5, Appendix A). For example, S-methyl cysteine sulfoxide, a dietary metabolite found in several vegetables, including cabbages, leeks, garlic, and onions [43], was strongly decreased in the first eight days. Other markers of dietary changes included carotene diols (decreasing), which is naturally sourced from peppers [44], and several increased flavouring agents, such as erythritol, maltol sulphate, and vanillic alcohol sulphate. Other flavouring agents, such as catechol sulphate and derivatives, showed both increases and decreases over time (catechol sulphate decreasing, 4-methylcatechol sulphate increasing). Benzoate, a widely used antimicrobial agent and food preservative [45], and its downstream metabolite hippurate were also increased over time. Notably, the increase in hippurate (Beta = 0.048, FDR = 0.009) was much weaker compared to benzoate (Beta = 0.091, FDR = 2.95 × 10^−12^, Figure 5).

### 2.6. COVID-19 Related Hospitalisation Impacts Host–Microbiome Co-Metabolism

Beyond the metabolomic changes including markers of environmental exposure, drug metabolism, and dietary metabolites, metabolites related to host–microbiome interactions were prominently placed among the significant metabolites (Figure 6, Appendix A). Several secondary bile acids, which are synthesised from primary bile acids via microbial conjugations, were accumulating in the serum within the first eight days of hospitalisation. All major secondary bile acid forms were increased, including deoxycholate, lithocholate, ursodeoxycholate acid, and their glycated/taurinated derivatives. Notably, several primary bile acids were also increasing, namely chenodeoxycholate conjugates. Cholate and its conjugates did not significantly alter in their concentrations. These results suggest an altered bile acid host–microbiome co-metabolism. Tryptophan host–microbiome metabolism was also affected. Although tryptophan concentration changes could not be analysed in the regression analyses due to tryptophan being depleted in more than 20% of all samples, tryptophan depletion could be inferred from its downstream metabolites, such as the microbially produced 3-formylindole [46] (decreased) and the increase of the microbiome-mediated tryptophan degradation products [47], including indolelactate and indoleacetate. Other human degradation products of tryptophan, namely anthranilate and methoxykynurenate, were also increased, indicating, together with the found overall tryptophan depletion, a higher turnover of tryptophan. In summary, we observed changes in serum concentrations of bile acids and tryptophan degradation products in newly hospitalised COVID-19 patients, indicating changing interactions between the host and gut-microbiome during hospitalisation. 

### 2.7. Indicators of Changed Physiological Functioning Are Reflected in the Serum Metabolome Trajectories

The serum metabolome also contained several markers related to physiological function in progressing COVID disease trajectories (Figure 7, Appendix A). For example, the top hit that consistently changed in this study was carboxyethyl-GABA (increased), which is a GABA derivative detected in human cerebrospinal fluid [48] and a faecal metabolite [49]. Other top hits in our study included a stark increase of fibrinopeptide A and B, which are components of fibrin [50], a major component in the coagulation cascade, which stops bleeding after vessel trauma [51]. Cholesterol sulphate was another increased top hit with known coagulation-inducing properties [52]. Additionally, several bilirubin degradation products were increasing, hinting at accelerated heme degradation. Although no conclusions can be made on how useful these compounds are in predicting thrombosis, these results do warrant further investigations into the clinical relevance of fibrinopeptides, cholesterol sulphate, and bilirubin degradation products in predicting thrombosis. Another compound of potential clinical interest was 3-methylglutaconate (increased), which is a known marker of metabolic acidosis [53]. Metabolic acidosis is a potential complication of severe COVID-19 [54], as the lung is one of the key regulators of blood pH value. The first week of hospitalisation also saw alterations in vitamin status with pyridoxal (vitamin B6) and retinol (vitamin A) decreasing, while tocopherols (vitamin E) displayed an inconsistent pattern (alpha-tocopherol increasing, beta/gamma-tocopherol decreasing). Of relevance in this context, 2-methyl-ascorbic acid, a vitamin C metabolite, was strongly increasing. While it is unclear whether these changes were caused by COVID-19 or changed nutrition during hospitalisation, it shows that the vitamin status was affected in hospitalised COVID-19 cases in a replicable pattern across three independent hospitals. In summary, the metabolomic changes point towards broad alterations in disease-relevant physiological processes, although, given the design of the study, it remains unclear whether these results were caused by COVID-19 or by the hospitalisation.

### 2.8. Metabolomic Results Reveal Potential Markers of Metabolic Reprogramming in Hospitalised COVID-19 Cases

The metabolomic analyses also revealed evidence for SARS-CoV-2-induced metabolic reprogramming (Figure 8 and Figure 9, Appendix A). Various lipid metabolites were changed after one week of hospitalisation, including several sphingomyelins, glycerophosphorylcholine, sphinganine-1-phosphate, and phosphatidylethanolamine (Figure 8, Appendix A). Although no consistent trend was found in metabolites in any of these classes, these changes in lipid metabolism were consistent with previous findings that SARS-CoV-2 rewires lipid metabolism to promote its replication spread [55]. Other possible markers of metabolic reprogramming were altered concentrations of amino acids (Figure 8 and Figure 9, Appendix A), such as valine (decreasing), arginine (decreasing), lysine (decreasing), histidine (decreasing), glycine (increasing), and serine (increasing). 

We found indications of SARS-CoV-2-induced metabolic reprogramming in the urea and TCA cycle. The measured substrates of the urea cycle, including citrulline and ornithine, decreased over time, whereas the products of the urea cycle, namely fumarate and urea, increased (Figure 9A,B). This result could be interpreted as higher fluxes through the urea cycle after one week of hospitalisation. Similar to the urea cycle, several TCA cycle metabolites decreased, including citrate, isocitrate, and aspartate (Figure 9A,B, Appendix A). Interestingly, glucose also decreased during hospitalisation (Appendix A). In conclusion, these results revealed metabolomic patterns, which may relate to processes of metabolic programming over the course of a viral infection. While these results cannot be directly linked to disease progression, they do suggest a distinct set of SARS-CoV2-influenced metabolite trajectories in the serum of newly hospitalised COVID-19 patients.

### 2.9. Metabolite Trajectories during Hospitalisation Were Dependent on Disease Severity

The results described above refer to changes in metabolite concentration regardless of the severity of COVID-19. In the next step, however, we investigated whether the changes in the serum metabolomes during hospitalisation were dependent on disease severity. Note that this analysis was confined to the Ticino cohort, as this was the only hospital having samples for severe and moderate COVID-19 cases. We performed mixed-effect regressions as above, introducing, however, the severity of COVID-19 (binary: moderate vs. severe) as a predictor of interest as well as an interaction term between the time variable and the COVID-19 severity. While only three annotated metabolites (caprylate, methylsuccinate, and xanthurenate) had significantly different serum concentrations between the two conditions (Appendix A) across all time points after correction for multiple testing by correcting for the false discovery rate, 17 metabolites had a significantly different time course during the first eight days in hospital dependent on the severity of COVID-19 (Figure 10, Appendix A). It is noteworthy that glycerophosphorylcholine, a glycophospholipid degradation product increasing generally in serum during the first week of hospitalisation (Appendix A), was increasing in severe COVID-19 cases and decreasing in moderate cases (Figure 10), making it a potential marker for monitoring disease progression. Cysteine-S-sulphate, another top hit, was decreasing much more steeply in moderate cases compared to severe cases (Figure 10). Interestingly, cysteine-S-sulphate is a metabolite known to be a biomarker of sulphite oxidase insufficiency [56], a rare disease leading to severe neurological dysfunction. Importantly, cysteine-S-sulphate is a very potent N-methyl-D-aspartate receptor agonist [57]. Some of the metabolites with severity-dependent trajectories were also found to change due to hospitalisation (Appendix A). Notably, several bilirubin degradation products were also among the metabolites with altered trajectory in severe COVID-19, as well glycochenodeoxycholate glucuronide. Taken together, we found various compounds that may serve as biomarkers for disease progression.

In a further step, we analysed whether trajectories of metabolites were altered in cases that died of COVID-19 and thus could serve as early biomarkers for COVID-19 mortality. Here, we combined the samples from Geneva and St. Gallen, since the number of deaths in the Ticino cohortwas not sufficient for statistical analysis. Once again, we performed mixed-effect regressions, introducing this time death by COVID-19 (binary: died of COVID-19 vs. survived) as a predictor of interest as well as an interaction term between the time variable and COVID-19 survival. However, the analysis did not reveal any biomarker after correction for multiple testing, hinting at missing statistical power (Appendix A). Taken together, while the analyses could identify biomarkers of severe COVID-19 with severity-dependent time trajectories, we could not identify biomarkers related to survival in this string of analysis.

### 2.10. Metabolite–Metabolite Relations Are Affected by Disease Severity and Disease Outcome

Next, we analysed the effect of disease severity on the bivariate distributions of all pairs of metabolites in the Ticino cohort. To this end, we calculated mixed-effect linear regressions as before, including, however, each metabolite and a metabolite-severity interaction term into the regressions as predictors. We then tested the interaction term on significance. This resulted in 901 × 901 = 811,801 tests, and we corrected *p*-values accordingly to account for multiple testing via Bonferroni correction. This analysis tests whether the statistical relation between two metabolites is influenced by the severity of the disease. We identified 14 metabolite–metabolite pairs where the statistical relation was significantly influenced by disease severity after correction for multiple testing (Figure 11A). For example, the correlation between N-acetyl-glutamate and cinnamoylglycine was clearly positive in severe COVID-19 cases, while being negative in moderate cases. For pyruvate and thymolsulfate, no correlation was found in moderate cases, while a positive association was detected in severe cases. The widespread alterations in bivariate metabolite–metabolite distributions depending on COVID-19 severity highlight the systemic changes due to COVID-19.

Finally, we tested whether people who died had different metabolite–metabolite dependencies, pooling the samples from St. Gallen and Geneva and following the same procedure as above with the difference that the interaction term was now defined as a metabolite–death interaction term. We found two altered metabolite–metabolite relations after correction for multiple testing (Figure 11B). Interestingly, the correlation between sulphate and 1-methyl-myristoylglycerol was far stronger in severe COVID-19 cases that later died compared to severe, surviving COVID-19 cases. As we found many sulphated metabolites to be altered due to hospitalisation, these results together may hint at sulphation processes being of relevance in COVID-19. In conclusion, while individual bivariate metabolite–metabolite distributions are difficult to interpret, our analyses provided ample evidence of altered metabolite–metabolite distributions in connection to disease severity and disease outcome.

## 3. Discussion

In this study, we described how the serum metabolomes of COVID-19 patients changed in the first week after hospitalisation in three independent Swiss hospitals. The repeated measurement design allowed us to assess the trajectory of the serum metabolome in 71 patients and revealed wide-spread metabolomic alterations during hospitalisation in the first eight days. In total, after meta-analysing the results, we found 448 metabolites consistently changing over time out of the 901 analysed compounds. This list covered all measured biochemical classes and showed surprising consistency in the detected profiles across the three hospitals. To the best of our knowledge, this is the first study integrating metabolome data from three independent hospitals in the context of hospitalisation. 

In the following, we will highlight several aspects of our results that could deserve further investigation for their potential clinical importance. In particular, we will discuss (1) metabolomic changes in relation to the hospital exposome, including dietary behaviour and drug metabolism, (2) metabolomic changes related to host–microbiome interactions, (3) potential markers of COVID-19-related pathophysiology, and (4) metabolomic changes that deliver potential markers of viral reprogramming of the host metabolism. Together, these results reveal important patterns that, while not being directly translatable in terms of clinical outcomes, can point towards the processes that are responsible for adverse COVID-19 trajectories or physiological resilience to COVID-19.

### 3.1. Metabolomic Changes Related to the Hospital Exposome

We found clear indications of changes in environmental exposure due to hospitalisation, covering so-called forever-chemicals, including perfluorooctanoate (decreased). Perfluorooctanoate has an estimated half-life between 0.5 and 1.5 years in the blood [58]. However, unaccounted background exposures have been known to result in widely varied perfluorooctanoate half-life estimates [59]. Although the timeframe of this study is only one week, our results similarly seem to show more rapid decreases in PFOA concentrations compared to what would be expected based on the half-life estimates, with average decreasing PFOA concentrations in one week of 16%, 26%, and 37% in Geneva, St. Gallen, and Ticino, respectively. These results seem to suggest a faster decrease in serum PFOA when changing environments. Future studies on this topic, however, would need to test this hypothesis. Our results also showed indicators of decreased usage of cosmetic and body care products, highlighting the breadth of metabolic changes that can be detected via untargeted metabolomics. Importantly, none of the markers of environmental exposure showed evidence of a disease-dependent trajectory in blood. As all severe patients in our cohorts were situated in the ICU and all moderate patients were situated in the hospital ward, this result also suggests that we did not find environmental exposures specific to the hospital ward or the ICU that influenced disease severity. Nevertheless, we believe that the topic of environment-disease interactions in the hospital deserves further considerations in future studies dealing with hospitalised patients. 

Although no data on medication usage were integrated in this work, the serum metabolome provided information, on which drugs were administered and how these drugs were degraded. For example, changes in acetaminophen metabolism were reflected in an increased concentration of glucuronide conjugated degradation products of acetaminophen while sulphate conjugates decreased. This decrease might be explained by the longer half-life of glucuronide conjugates [60]. An alternative explanation of this result is that the increase of glucuronide conjugates was due to patients in all locations having an overweight BMI (BMI > 25), as a higher BMI has been associated with an enhanced glucuronide conjugation in overweight individuals [61].

Our results also included several markers of changed dietary behaviour, in particular hinting at decreased intake of compounds commonly found in vegetables and an increased intake of food additives and food preservatives. These changes in dietary markers hint at a reduced nutrient content during the hospital stay, which would be in line with previous findings that reported 66.7% of patients in the ICU and 23.7% of the moderate patients in the hospital ward to be malnourished [62]. It should be noted that the measured changes in these dietary outcomes are likely driven by patients in the ICU, given the fact that ICU patients might have needed enteric or intravenous nutrition. The statistical power to test differences in dietary changes in moderate and severe patients, however, was not enough to derive clear inferences due to the low number of moderately ill patients. Future studies on this subject should investigate how differences in ICU nutrition and the hospital ward diet may influence the patient’s metabolism and physiology.

In conclusion, although hospital exposome-related compounds could not be linked with disease severity or mortality, we believe that the measured exposome-related serum changes are a relevant step towards a better and more holistic understanding on exposome–disease interactions.

### 3.2. Metabolomic Changes Related to Host–Microbiome Interactions

We also found markers of diet–microbiome interactions in changes in benzoate metabolism. The human gut-microbiome is a known modulator of benzoate degradation [63] and has evolved pathways to protect against the normally antimicrobial properties of benzoate [64]. The benzoate degradation product hippurate was also increased, but much less then benzoate. One possible explanation might be that changed microbiome compositions, which are known to occur in COVID-19 [65], reduced the importance of benzoate degradation via hippurate. While unclear in their clinical importance, these results indicate that hospitalisation has strong effects on the human serum metabolome due to changes in diet and environment. Furthermore, we also found markers of changing host–microbiome co-metabolism, as primary bile acids had an increased turnover rate and secondary bile acids were accumulating in the serum. These findings are in line with earlier findings on gut-microbiome compositions, as both in-hospital disease progression and disease severity have been found to correlate with higher abundances of pathogenic and opportunistic bacteria and lower abundances of beneficial bacteria [65,66].

### 3.3. Potential Markers of COVID-19 Related Pathophysiology

Besides markers of changing host–gut microbiota interactions, we found various other domains of metabolism that may be related to physiological changes and disease progression. For example, the top hit in this study, carboxyethyl-GABA (increased), is known to promote cell proliferation and migration in mouse fibroblasts [67], which are cells that function as support in the structural integrity of the intercellular matrix and play an important role in wound healing. The increased carboxyethyl-GABA concentrations over time led us to hypothesise that carboxyethyl-GABA could be a possible marker of interstitial pulmonary fibrosis, a group of diseases marked by the pathological healing of lung tissue and pathological fibroblast behaviour. Interstitial pulmonary fibrosis is associated with lung injury and is common in severe COVID-19 disease trajectories [68]. We also found indicators of increased coagulation activity (Figure 7, Appendix A). Fibrinopeptide A and B increased during hospitalisation. These compounds are released in the coagulation cascade from fibrinogen in the formation of fibrin [69], which forms a matrix around vessel lesions and captures platelets to produce blood clots [51,70]. Another indicator of increased coagulation was the higher level of cholesterol sulphate during hospitalisation. Cholesterol sulphate is present in platelet membranes and promotes platelet adhesion and clotting [71]. Serum fibrinopeptide and cholesterol sulphate concentrations could be potential markers to assess the risk of a patient developing intravenous thrombosis, which is a common complication in severe COVID-19 patients [72]. Anticoagulation drugs, such as low molecular weight heparins [73], could be given to patients at high risk of developing intravenous thrombosis.

Our results also showed increased concentrations of several bilirubin degradation products during hospitalisation (Figure 7, Appendix A). Bilirubin is a waste product from the breakdown of red blood cells and is metabolised in the liver and subsequently degraded by microbes in the colon [74]. Previous findings have linked increases in serum bilirubin and its degradation products in COVID-19 patients to decreased liver function [33], but increased bilirubin degradation could also be explained by SARS-CoV-2-induced breakdown of red blood cells [75]. 3-methylglutaconate was proposed as a potential marker of severe COVID-19 outcomes due to its properties related to the prediction of metabolic acidosis [76]. Although no conclusions can be made on how useful these compounds are in predicting thrombosis, these results do warrant further investigations into the clinical relevance of these compounds in predicting lung injuries and thrombosis.

### 3.4. Potential Markers of Viral Reprogramming of Host Metabolism

Our results also included possible signals of SARS-CoV-2-induced metabolic programming of the host. Notably, the urea cycle was found to be dysregulated, with urea cycle substrates arginine, citrulline, and ornithine decreasing over time while its products accumulated. Interestingly, these results confirm metabolic modelling efforts on SARS-CoV-2-driven metabolic reprogramming of the host, which also found decreased fluxes of citrulline and ornithine and increased fumarate fluxes when comparing mild and severe COVID-19 patients against non-infected individuals [28]. Besides a higher flux through the urea cycle, we observed that citrulline decreased more over time than ornithine (Figure 9). Although these differences in the regression slopes could be explained by processes not captured in this study, this observation could also indicate that the increased urea production (Figure 9) from arginine resulted in a decreased production of nitric oxide from arginine. However, as nitric oxide was not measured in our data, no conclusions on SARS-CoV-2 influences on nitric oxide productions can be made. SARS-CoV-2 has been hypothesised before to benefit from a decrease in nitric oxide production, as nitric oxide is known to inhibit the early steps in the replication of the original SARS-CoV virus [77]. At the same, an increased production of ornithine could promote viral replication as it is an important precursor of several polyamines, which play important roles in viral replication [78]. These dysregulations hint at a worsening COVID-19 progression as urea cycle dysregulation and reduced importance of nitric oxide production from arginine have also been found when comparing healthy individuals with moderate and severe COVID-19 patients [79].

Indications of SARS-CoV-2-induced metabolic programming were also found in decreased concentrations of several TCA cycle metabolites, including aspartate, citrate, and isocitrate, and in a decrease of serum glucose concentrations. Although no causal inferences can be made from our results, it is likely that SARS-CoV-2 played a role in these metabolic changes as SARS-CoV-2 is known to increase TCA cycle activity by promoting the cellular uptake of glucose [27]. The dysregulation of TCA cycle metabolites in our cohorts agrees with previous results where healthy individuals were compared to mild and severe COVID-19 patients, which might again suggest a worsening of COVID-19 disease progression. To summarise, we found drastic changes in urea and TCA cycle metabolite concentrations in the first week after hospitalisation. These results agreed with previous findings that associated urea and TCA cycle metabolites with COVID-19 disease severity [79]. Therefore, we hypothesise that urea cycle and TCA cycle metabolites could be possible prognostic markers of disease progression.

We also found hints of metabolic reprogramming by testing bivariate metabolite–metabolite trajectories against disease severity and disease outcome. Although our results could be confounded by the altered prevalence of disease-severity-associated comorbidities, the altered metabolite trajectories still present potential biomarkers for disease progression and disease monitoring. The top hit was glycerophosphorylcholine, a phospholipid degradation product, which had a far steeper increase in the serum in severe cases in comparison to moderate cases. Glycerophosphorylcholine is a known marker of COVID-19 disease severity [22,23,80]. Interestingly, phospholipids in general are known to be beneficial for viral replication, as they are important constituents of lipid droplets and lipid rafts, which play major roles in the viral lifecycle [81,82]. These results align with previous insights into SARS-CoV-2 rewiring of phospholipid metabolism [22,23,80], thus making serum glycerophosphorylcholine trajectory a potential marker for disease severity.

### 3.5. Strengths and Limitations

The main strength of the study is the possibility of meta-analysing the results of three independent cohorts from three different hospitals with repeated metabolome measurements. This study design enabled finding consistent metabolomic changes across the three cohorts, while the use of the untargeted metabolomics approach made it possible to access a wide range of serum metabolites. Thus, we could analyse not only the main players in the human metabolism, such as amino acids or lipids, but also markers of environmental exposure, shedding a light on the complex and intertwined changes in the metabolome during hospitalisation. The study, however, has some important limitations. It is of observational design, and as such causal inferences are generally impossible, due to unmeasured confounding factors, making interpretations of individual results difficult. As a consequence, our findings cannot be attributed to COVID-19 since hospitalisation also leads to changes in environmental exposure and behavioural changes, e.g., physical activity, medication, and diet, which all influence human metabolism. Moreover, the limited sample size in the cohorts, while being enough to detect the vast and drastic changes in the trajectories, may not have been sufficient to detect all of the relevant effects present. Part of the analyses included utilising interaction terms, where it is known that large sample sizes are needed for adequate statistical power. Therefore, it is conceivable that when it comes to metabolite–metabolite relations, many true effects were not detected, delivering an incomplete picture. A further limitation is that we did not analyse the effect of comorbidities on metabolomics trajectories, as this was beyond the scope of this work.

### 3.6. Conclusions

In conclusion, our work revealed the drastic effects of COVID-19 hospitalisation on the human serum metabolome. We found markers related to hospitalisation, including environmental, dietary, and drug metabolism. Our results also included possible markers of changed lung injury, including carboxyethyl-GABA and fibrinopeptides. Furthermore, we proposed urea cycle metabolites, TCA cycle metabolites, and glycerophosphorylcholine as potential markers of COVID-19-induced metabolic reprogramming and disease progression. We hope that the reported results prove to be fruitful and helpful for the interpretation of metabolome data sampled within hospitals in general.

## 4. Methods

### 4.1. Study Cohorts

In order to analyse serum metabolome trajectories in COVID-19 patients during hospitalisation, three prospective cohorts were selected across different locations across Switzerland. A total of 71 patients were recruited in total, with 29 patients from Geneva, 22 patients from St. Gallen, and 20 patients from Ticino (Table 1). The Geneva cohort originally included 30 patients; however, one patient was dropped as they did not have information on mortality and sex. Patient recruitment started in August 2020. The selection criteria included that the study participants were adults ≥18 years and were admitted to the hospital ward or ICU due to PCR-confirmed SARS-CoV-2 infection. Finally, participants were only included if they or their representatives had signed an informed consent form. This study was approved by a local ethics committee (EKOS 20/058). The final cohorts included moderate and severe cases in Ticino and only severe cases in Geneva and St. Gallen. Moderate cases were defined as PCR-confirmed SARS-CoV-2-infected patients with symptoms of pneumonia, fever, and respiratory tract problems. Severe COVID-19 patients had all the symptoms of moderate cases, but also needed to have a respiratory rate of ≥30 breaths per minute and an oxygen saturation of ≤93% when breathing ambient air or having a PaO_2_/FiO_2_ below 300 mmHg. Patients that did not meet these requirements but needed ventilator support were also classified as severe COVID-19 patients. Moderately ill patients were all staying in a hospital ward, whereas severe patients were all situated in the intensive care unit. 

### 4.2. Sample Collection and Treatment

All patients that were willing to partake in this study and passed the inclusion criteria had a first blood sample collected within 24–48 h after hospitalisation. Subsequent samples were generally taken every week after hospitalisation (Appendix A). Collected samples were immediately stored at −80 °C until processing. Sample preparation was performed by Metabolon^TM^ using the automated MicroLab STAR^®^ system from Hamilton Company^TM^ (Reno, NV, USA). Proteins were removed by dissociating small molecules bound to protein or trapped in the precipitated protein matrix. Chemically diverse metabolites were then recovered by precipitating proteins with methanol under vigorous shaking for 2 min (GenoGrinder 2000^®^, Glen Mills Inc., Clifton, NJ, USA), followed by centrifugation. The samples were briefly placed on a TurboVap^®^ (Zymark Corp, Portland, OR, USA) to remove the organic solvent, after which they were stored overnight under nitrogen before preparation for analysis. 

### 4.3. Metabolomics

Untargeted metabolomics data were generated from patient serum samples using the Metabolon^TM^ using the HD4 platform. Waters ACQUITY ultra-performance liquid chromatography (UPLC) was performed with a Thermo Scientific (Waltham, MA, USA) Q-Exactive high-resolution/accurate mass spectrometer interfaced with a heated electrospray ionization (HESI-II) source and Orbitrap mass analyser, which operated at a mass resolution of 35,000. Before analysis, the serum sample extract was dried and reconstituted in four separate solvents compatible with each of the four used methods. The first aliquot was analysed in acidic positive ion conditions, which were chromatographically optimised for hydrophilic compounds. The second aliquot was analysed in acidic positive ion conditions, which were chromatographically optimised for more hydrophobic compounds. The third aliquot was analysed using a separate dedicated C18 column in basic negative ion optimised conditions. Analysis of the fourth aliquot was conducted via negative ionization after elution from a HILIC column (Waters UPLC BEH Amide 2.1 × 150 mm, 1.7 µm). A gradient was used consisting of water and acetonitrile with 10 mM ammonium formate and a pH of 10.8. The mass spectrometry (MS) analysis was performed in an alternating manner between MS and data-dependent MS^n^ scans using dynamic exclusion. There were slight variations between the methods, but the scan range covered 70–1000 *m*/*z*. 

The serum samples were analysed by the Metabolon platform in two batches. Batch effects were mitigated by running 12 anchor samples from healthy volunteers in both Metabolon runs. The second batch was rescaled to be on an identical scale as the first batch. First, the anchor sample metabolite ratios were computed as follows: $ratio_{x,y}=metabolite{\left(x,y\right)}_{batch\;1}/metabolite{\left(x,y\right)}_{batch\;2}$, where x corresponds to the anchor sample and y corresponds to the metabolite area under the peak value. The second batch was then rescaled by multiplying the metabolite values with the median of twelve anchor sample ratios. If a metabolite was not measured in more than half of the anchor datasets, scaling was not performed for that metabolite and the metabolite was excluded from our analysis. After batch scaling, missing values were imputed with the minimum value measured for a metabolite value.

Metabolites that were not present in at least 20% of all samples in the three cohorts were removed. Data processing and sample removal were performed using the Tidyverse [83] software suite in the R programming language version 4.2.2. 

### 4.4. Analyses of Differential Metabolite Trajectories over Time

Metabolite trajectories and metabolite depletion trajectories were analysed by performing linear and logistic mixed-effect regressions with, respectively, the log-transformed metabolite concentrations as the outcome. Both regression models used the number of days after hospitalisation as the predictor of interest and included age, sex, BMI, and death due to COVID-19 as control variables. The individual patient was used as the random intercept. Both regression models were performed on each of the 901 selected metabolites and for each location.

Next, we meta-analysed the metabolite and metabolite depletion regressions for each of the 901 metabolites. We treated the regression coefficients ${\hat{\beta }}_{i},\;i=1,\ldots ,\;3\;$of the standardised metabolite concentrations for each of the three studies as the individual effect estimates we wanted to summarise. Given the sample estimates ${\hat{\sigma }}_{i}^{2}$ of Var(${\hat{\beta }}_{i})$ and assuming that the individual effect estimates were fixed outcomes of the three studies, we calculated the overall effect estimates $\bar{\beta }$ using a fixed-effects model [84] and the inverse variance method for each metabolite. We quantified heterogeneity using Cochran’s Q-test for each of the 901 metabolites. After FDR correction of the *p*-values of the overall effect estimates, for 545 metabolites, the overall effect remained significant (FDR < 0.05). After that, we filtered out the metabolites of the total 901 that were significant (FDR < 0.05) according to Cochran’s Q-test, resulting in 448 metabolites.

Additionally, we also investigated time-dependent trajectories in the number of samples where a metabolite was depleted, i.e., metabolites with concentrations below the detection limit in a sample. We ran logistic mixed-effect regressions for each location with the binary metabolite detected/not detected as the outcome and the days after hospitalisation as the predictor of interest. We again controlled for age, sex, and BMI, with random intercepts for the individuals, and meta-analysed the results as we did with the metabolite concentration trajectories. However, due to most metabolites having too few depleted samples in one or more locations, meta-analysed results could only be obtained for 22 metabolites. Of these 22 metabolites, five were nominally significant, which were all already identified in the regressions on the metabolite concentration trajectories. Due to the uninformative results from this analysis, it was dropped from the reported results. 

The regression analyses and meta-analyses were performed using the following software packages in R (v.4.2.2). Metabolite trajectories were calculated using the PLM package [85]. The lme4 package [86] was used to perform the logistic mixed-effect regressions on the metabolite depletions and to calculate the 95% confidence intervals. All calculations for the meta-analysis were performed using the metafor package [84]. 

### 4.5. Analyses of Time-Dependent Metabolite Trajectories against Disease Severity and Outcome

Potential biomarkers that could differentiate between moderate and severe cases were investigated by performing mixed-effect regressions on the log-transformed metabolite concentrations (response) against the disease severity (binary: moderate vs. severe) as the predictor of interest, while controlling for age and sex and the interaction between disease severity and the number of days after hospitalisation. The *p*-values of the regression coefficients were corrected for the false discovery rate with an alpha value of 0.05. All samples between the first and ninth day after hospitalisation were included. This analysis was only conducted for Ticino, as it was the only cohort with both moderate and severe patients. In a second analysis, we again investigated potential biomarkers, but now between severe surviving and fatal cases in a combined cohort from St. Gallen and Geneva. Ticino was excluded as it did not include enough fatal cases (two fatal cases) for any useful statistical analyses. Mixed-effect regressions were performed on the log-transformed metabolite concentrations (response) against the binary: died of COVID-19 vs. survived as the predictor of interest. This time, we controlled not only for age and sex, but also for the location and the interaction between time and COVID-19 mortality. We again corrected for the false discovery rate with an alpha value of 0.05. 

### 4.6. Pathway Enrichment Analyses

Pathway enrichment analysis was performed on MetaboAnalyst 5.0 [38]. The HMDB IDs of the list of significant metabolites were uploaded to the MetaboAnalyst website, after which the metabolites were mapped onto the MetaboAnalyst database. A total of 289 metabolites could be mapped. Enrichment analysis was then performed against the KEGG *Homo sapiens* reference pathway library [39]. The enrichment impact and *p*-values were calculated with the Fisher exact test (hypergeometric test) using a relative-betweenness centrality. 

### 4.7. Disease-Severity-Dependent Metabolite–Metabolite Interactions

Associations between bivariate metabolite–metabolite distributions and disease severity were investigated by performing mixed-effect linear regressions for all 901 × 901 = 811,801 metabolite pairs with one metabolite as response and predictors for a second metabolite and the metabolite-severity interaction term. Control variables were added for age and sex. The resulting outcomes were corrected via Bonferroni correction to account for multiple testing. The relationship between metabolite–metabolites distributions and disease severity were only tested in Ticino as this was the only location with both moderate and severe COVID-19 patients. Bivariate metabolite–metabolite distributions were associated with COVID-19 mortality in the same manner as outlined earlier, with the difference that an interaction term for metabolite concentration and disease mortality was used as the predictor of interest.

## Figures and Tables

**Figure 1 metabolites-13-00951-f001:**
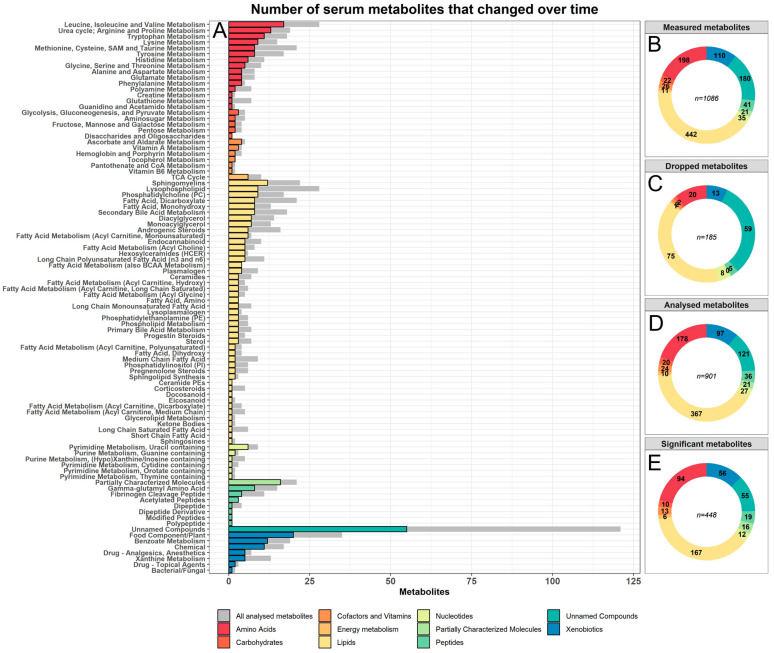
Distributions of serum metabolites by biochemical family. (**A**) Overview of the number of significantly and consistently changed metabolites per pathway in the first eight days after hospitalisation. Each pathway was coloured and sorted by their biochemical family. The grey bars represent the total number of metabolites analysed in each pathway. Panel (**B**–**E**) show the distributions of metabolites per biochemical family, respectively, for all measured metabolites, all removed metabolites, all analysed metabolites, and all significant and consistently changed metabolites.

**Figure 2 metabolites-13-00951-f002:**
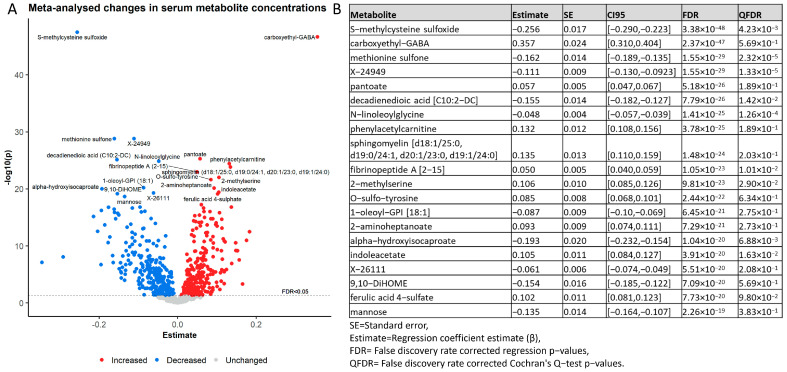
Overview of the top 20 most significantly changed serum metabolites. (**A**) Volcano plot of regression outcomes for all 901 analysed metabolites with the regression estimate on the x-axis against the −log10 transformed *p*-value on the y-axis. The red and blue dots represent all increased and decreased metabolites, respectively. The top 20 most-changed metabolites are labelled. (**B**) Summary of the regression results for the top 20 metabolites with the lowest FDR corrected *p*-values. In addition to the metabolite names, the standard errors (SE), the regression coefficient estimates (Estimate), and the 95% confidence intervals (CI95) are displayed. The FDR corrected *p*-values from the regression models are shown as FDR. The QFDR values represent the FDR corrected *p*-values obtained from the Cochran’s Q-test for quantifying the between-cohort heterogeneity.

**Figure 3 metabolites-13-00951-f003:**
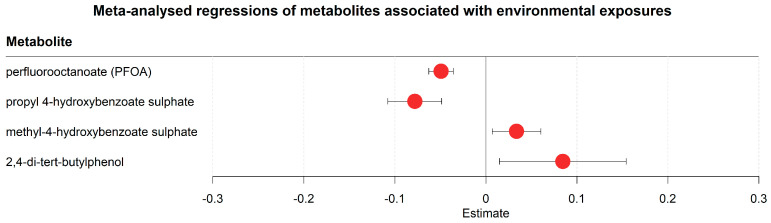
Meta-analysed regression outcomes of serum metabolites linked to environmental exposures. Forest plot of meta-analysed compounds linked to environmental exposures. The estimates, or regression coefficients, represent the pooled change in concentration over time in the three cohorts (see Section 4 for details). Negative estimates indicate decreased serum concentrations, while positive estimates indicate increased serum concentrations during hospitalisation. The displayed metabolites all changed consistently and homogenously between cohorts. All metabolites remained significantly changed after correction for the false discovery rate. The 95% confidence interval is given by the protruding lines from the metabolite estimate.

**Figure 4 metabolites-13-00951-f004:**
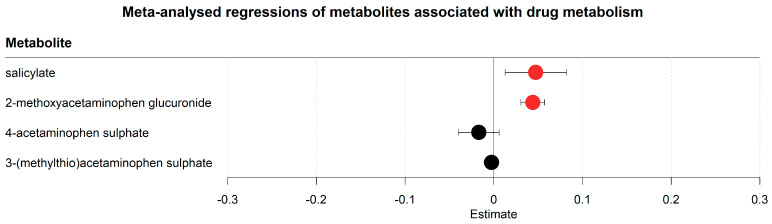
Meta-analysed regression outcomes of serum metabolites associated with drug metabolism. Forest plot of meta-analysed compounds associated with drug metabolism. The estimates, or regression coefficients, represent the pooled change in concentration over time in the three cohorts (see Section 4 for details). Negative estimates indicate decreased serum concentrations, while positive estimates indicate increased serum concentrations during hospitalisation. The displayed metabolites all changed consistently and homogenously between cohorts. Metabolites with red-coloured estimates remained significantly changed after correction for the false discovery rate. Black-coloured estimates indicate no significant change after multiple testing correction. The 95% confidence interval is given by the protruding lines from the metabolite estimate.

**Figure 5 metabolites-13-00951-f005:**
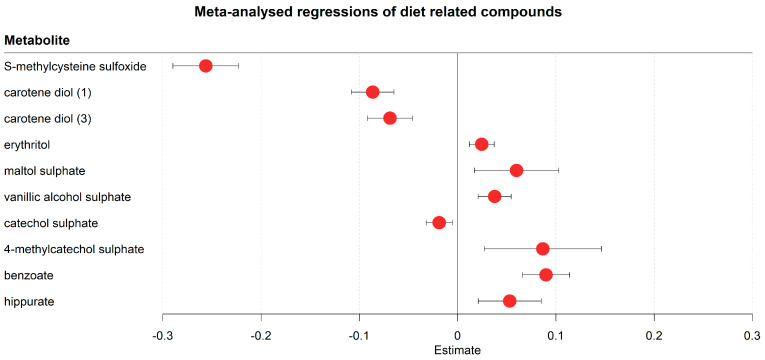
Meta-analysed regression outcomes of serum metabolites related to dietary behaviour. Forest plot of meta-analysed compounds related to dietary behaviour. The estimates, or regression coefficients, represent the pooled change in concentration over time in the three cohorts (see Section 4 for details). Negative estimates indicate decreased serum concentrations, while positive estimates indicate increased serum concentrations during hospitalisation. The displayed metabolites all changed consistently and homogenously between cohorts. All metabolites remained significantly changed after correction for the false discovery rate. The 95% confidence interval is given by the protruding lines from the metabolite estimate.

**Figure 6 metabolites-13-00951-f006:**
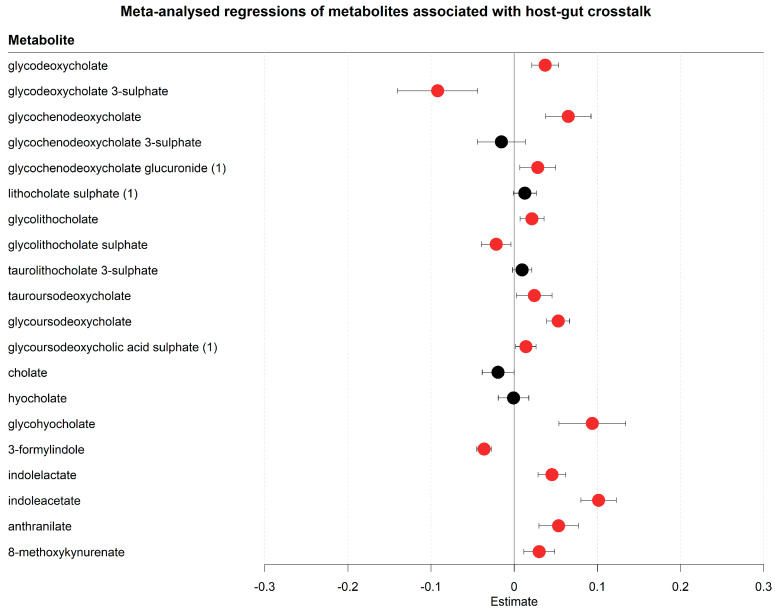
Meta-analysed regression outcomes of serum metabolites related to crosstalk between the host and gut-microbiome. Forest plot of meta-analysed compounds related to crosstalk between the host and gut-microbiome. The estimates, or regression coefficients, represent the pooled change in concentration over time in the three cohorts (see Section 4 for details). Negative estimates indicate decreased serum concentrations, while positive estimates indicate increased serum concentrations during hospitalisation. The displayed metabolites all changed consistently and homogenously between cohorts. Metabolites with red-coloured estimates remained significantly changed after correction for the false discovery rate. Black-coloured estimates indicate no significant change after multiple testing correction. The 95% confidence interval is given by the protruding lines from the metabolite estimate.

**Figure 7 metabolites-13-00951-f007:**
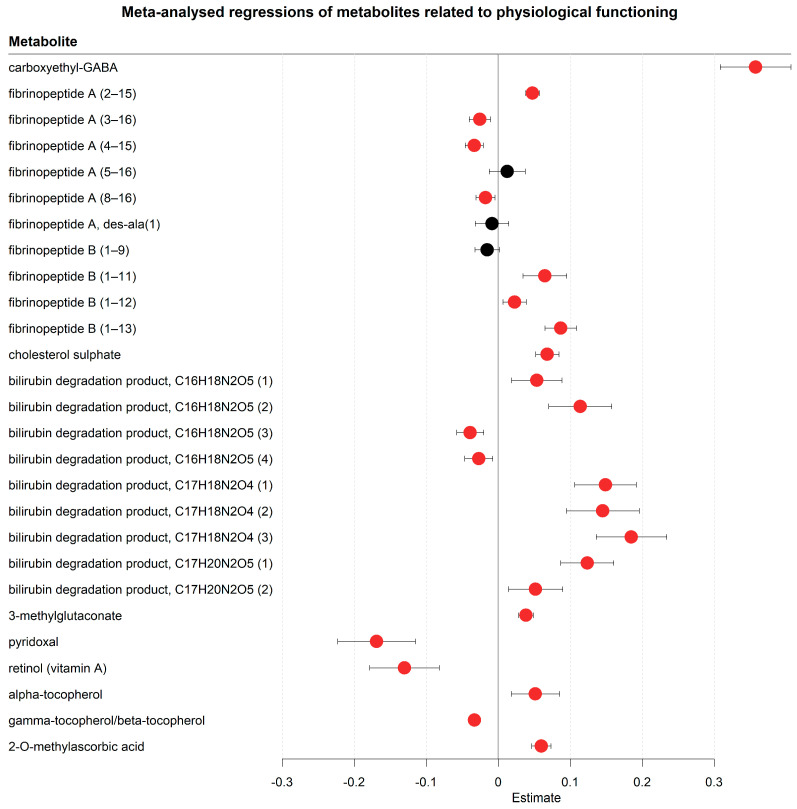
Meta-analysed regression outcomes of serum metabolites related to physiological functioning. Forest plot of meta-analysed compounds related to physiological functioning. The estimates, or regression coefficients, represent the pooled change in concentration over time in the three cohorts (see Section 4 for details). Negative estimates indicate decreased serum concentrations, while positive estimates indicate increased serum concentrations during hospitalisation. The displayed metabolites all changed consistently and homogenously between cohorts. Metabolites with red-coloured estimates remained significantly changed after correction for the false discovery rate. Black-coloured estimates indicate no significant change after multiple testing correction. The 95% confidence interval is given by the protruding lines from the metabolite estimate.

**Figure 8 metabolites-13-00951-f008:**
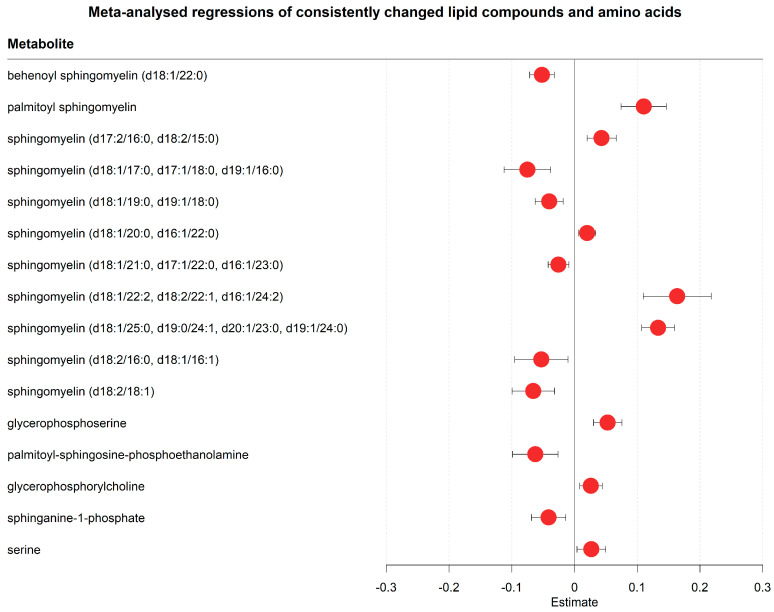
Meta-analysed regression outcomes of serum metabolites linked to SARS-CoV-2-induced metabolic reprogramming. Forest plot of meta-analysed compounds that are linked to SARS-CoV-2 induced metabolic reprogramming. The estimates, or regression coefficients, represent the pooled change in concentration over time in the three cohorts (see Section 4 for details). Negative estimates indicate decreased serum concentrations, while positive estimates indicate increased serum concentrations during hospitalisation. The displayed metabolites all changed consistently and homogenously between cohorts. All metabolites remained significantly changed after correction for the false discovery rate. The 95% confidence interval is given by the protruding lines from the metabolite estimate.

**Figure 9 metabolites-13-00951-f009:**
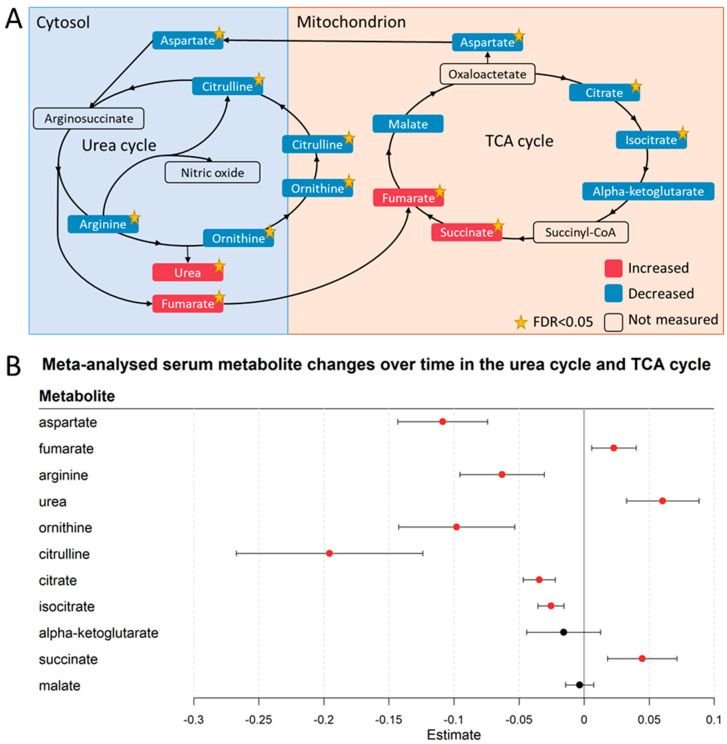
Metabolic reprogramming in the urea cycle and the TCA cycle. (**A**) Serum metabolic changes of the urea cycle metabolites and TCA cycle metabolites. Metabolites in red were increased in the serum after one week, whereas metabolites in blue were decreased. The yellow star indicates if the change was significant (FDR < 0.05) over time. All displayed metabolites were consistently changed across the three locations. (**B**) Forest plot of meta-analysed regression outcomes of the visualised urea cycle and TCA cycle metabolites in Figure 2A. The estimates, or regression coefficients, represent the pooled change in concentration over time in the three cohorts (see Section 4 for details). Negative estimates indicate decreased serum concentrations, while positive estimates indicate increased serum concentrations during hospitalisation. The displayed metabolites all changed consistently and homogenously between cohorts. Metabolites with red-coloured estimates remained significantly changed after correction for the false discovery rate. Black-coloured estimates indicate no significant change after multiple testing correction. The 95% confidence interval is given by the protruding lines from the metabolite estimate.

**Figure 10 metabolites-13-00951-f010:**
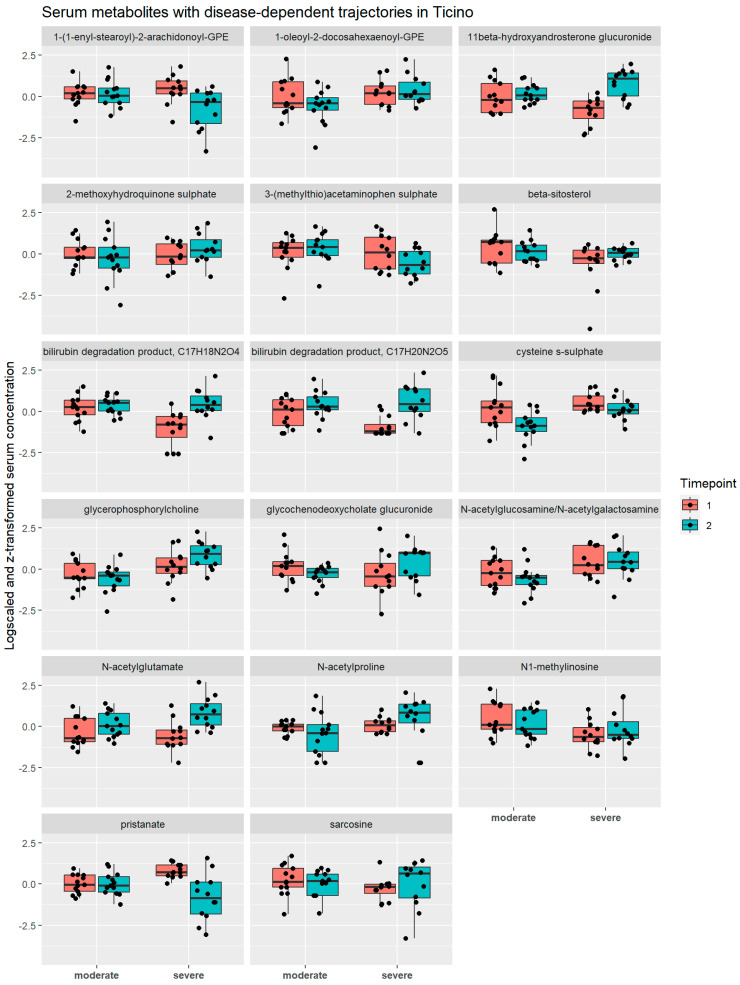
Overview of serum concentrations of metabolites with disease-dependent trajectories in the Ticino cohort. Boxplots of the log-transformed concentrations of metabolites with different serum trajectories in moderate and severe patients in Ticino. In each tile, comparisons are made between the first (salmon red) and second (turquoise) timepoint for the moderate cases (left two boxplots) and severe cases (right two boxplots). The black dots represent the individual concentration values.

**Figure 11 metabolites-13-00951-f011:**
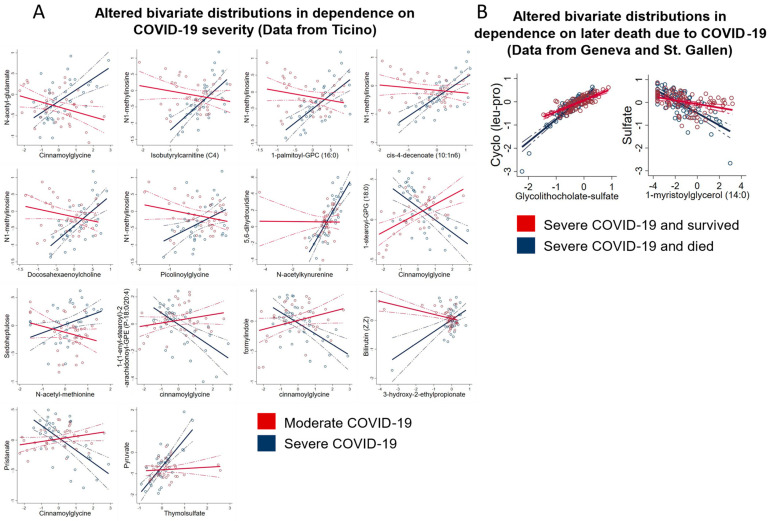
Altered bivariate metabolite distributions. (**A**) Altered bivariate distributions of metabolite–metabolite pairs in moderate (red) and severe (blue) COVID cases from Ticino. All shown metabolite–metabolite pairs differed significantly between moderate and severe COVID patients. (**B**) Significantly altered bivariate distributions of metabolite–metabolite pairs in severe COVID patients in Geneva and St. Gallen. Bivariate metabolite distributions of patients that survived COVID are shown in red, while bivariate metabolite distributions of patients that later died are shown in blue.

**Table 1 metabolites-13-00951-t001:** Summary of COVID-19 patient demographics from three Swiss hospitals. SD—standard deviation. BMI—body mass index.

**Geneva**	**Moderate**	**Severe-Survived**	**Severe-Fatal**
Analysed patients		14	15
Mean age (SD)		66.8 (9.6)	66.6 (8.82)
Female/male		5/9	3/12
Mean BMI (SD)		28.6 (6.2)	25.5 (4.2)
Analysed samples		28	30
**St. Gallen**	**Moderate**	**Severe-Survived**	**Severe-Fatal**
Analysed patients		11	11
Mean age (SD)		59 (10.6)	65.4 (8.98)
Female/male		1/10	2/9
Mean BMI (SD)		31.2 (6.1)	30.4 (3.7)
Analysed samples		22	22
**Ticino**	**Moderate**	**Severe-Survived**	**Severe-Fatal**
Analysed patients	11	7	2
Mean age (SD)	53.5 (9.11)	60 (8.56)	68.5 (3.54)
Female/male	5/6	2/5	1/1
Mean BMI (SD)	24.1 (4.4)	28.0 (2.0)	29.7 (5.7)
Analysed samples	22	14	4

## Data Availability

The datasets used in this study are available upon reasonable request. The data are not publicly available due to privacy. The used code for the metabolomics data processing and statistical analyses can be found publicly at https://github.com/ThieleLab/CovidMetabolomicsAnalysis. Further information and requests for resources should be directed to and will be fulfilled by the lead contact Ines Thiele.

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
