# Peer review of "The Effects of Hospitalisation on the Serum Metabolome in COVID-19 Patients"

_metabolites, 2023, doi:10.3390/metabo13080951_

Round 1

Reviewer 1 Report

The manuscript explored changes in metabolites of hospitalized COVID-19 patients and provided insightful information about serum metabolome changes associated with environmental and physiological factors through meta-analysis. Although the number of cases is relatively low for some of the analyses, the data presented in the manuscript is informative and enlightening for COVID-19 and metabolomics research. However, I have identified some minor issues in the manuscript, which are outlined below:

1. The 'SARS-COV-2' should be 'SARS-CoV-2', with the 'o' lowercased.

2. In line 106 and the title above Fig 6, 'host-gut interaction/crosstalk' should be corrected to 'host-gut microbiota crosstalk.'

3. In line 647, the sentence "This section is not mandatory but can be added to the manuscript if the discussion is unusually long or complex" seems confusing. It appears to have been mistakenly included from the manuscript template. Please review and clarify its relevance in the context of the study.

4. It is recommended to provide basic sample pretreatment and UPLC-MS information in the methods section or cite a published paper that describes the methodology. Specifically, please include details on how the blood samples were prepared for analysis, the type of chromatography employed (reverse phase or HILIC), and the ionization mode used (positive or negative).

5. The description of the number of patients recruited is confusing. In line 652, it is stated, "A total of 71 patients were recruited in total with 51 patients from Geneva, 37 patients from St. Gallen, and 42 patients from Ticino (Table 1)." In addition, in line 119, the author states that the samples from Ticino consisted of 20 severe and 22 moderate COVID-19 cases. However, these statements contradict the information provided in the context and Table 1. Please ensure consistency in reporting the number of cases throughout the manuscript.

Author Response

The manuscript explored changes in metabolites of hospitalized COVID-19 patients and provided insightful information about serum metabolome changes associated with environmental and physiological factors through meta-analysis. Although the number of cases is relatively low for some of the analyses, the data presented in the manuscript is informative and enlightening for COVID-19 and metabolomics research. However, I have identified some minor issues in the manuscript, which are outlined below:

The 'SARS-COV-2' should be 'SARS-CoV-2', with the 'o' lowercased.

>> All instances of 'SARS-COV-2' have been replaced by 'SARS-CoV-2'.

In line 106 and the title above Fig 6, 'host-gut interaction/crosstalk' should be corrected to 'host-gut microbiota crosstalk.'

>> 'Host-gut interaction/crosstalk' has been replaced by 'host-gut microbiota crosstalk'.

In line 647, the sentence "This section is not mandatory but can be added to the manuscript if the discussion is unusually long or complex" seems confusing. It appears to have been mistakenly included from the manuscript template. Please review and clarify its relevance in the context of the study.

>> We removed the sentence.

It is recommended to provide basic sample pretreatment and UPLC-MS information in the methods section or cite a published paper that describes the methodology. Specifically, please include details on how the blood samples were prepared for analysis, the type of chromatography employed (reverse phase or HILIC), and the ionization mode used (positive or negative).

>>Thank you for this important comment. The sample pretreatment is described in a new section in the methods with the name “Sample collection and treatment”. Blood sample preparation for analysis is explained in detail. Another new section entitled “Metabolomics” was added, describing in detail the metabolomics data generation including all UPLC-MS information. We specifically ensured that the type of chromatography and ionisation mode were given.

The description of the number of patients recruited is confusing. In line 652, it is stated, "A total of 71 patients were recruited in total with 51 patients from Geneva, 37 patients from St. Gallen, and 42 patients from Ticino (Table 1)." In addition, in line 119, the author states that the samples from Ticino consisted of 20 severe and 22 moderate COVID-19 cases. However, these statements contradict the information provided in the context and Table 1. Please ensure consistency in reporting the number of cases throughout the manuscript.

>> Thank you very much for pointing out the inconsistency, which we have now removed. Line 652 stated sample numbers for the three locations that included patients that were removed in our analyses. The sample numbers for Geneva, St. Gallen, and Ticino were updated to be in line with the correct sample numbers shown in Table 1.

Line 747: “homo sapiens” —> Please write it in italics, and h of homo in capitals.

>> Thank you for this correction. We corrected “homo sapiens” to “Homo sapiens” in the revised manuscript.

Please add the description of supplementary materials at page 23 (1), as well as the informed consent statement (2), data availability statement (3) and conflicts of interest (4).

>> Thank you for these useful comments. The description of figure S1 (1) is updated and completed and the informed consent statement (2), data availability statement (3) and conflicts of interest (4) are added.

Reviewer 2 Report

The authors aim to investigate how the hospital exposome, disease progression, and exposome-disease interactions influence physiological functioning, through the untargeted analysis of serum metabolomics of 71 hospitalised moderate and severe COVID-19 patients.

The methods sound correct for the investigation. Manuscript is clear and the language does not need revision.

However, the major limitation for this study lies in the clarity for clinicians. Indeed, physicians not expert in biochemical mechanisms would find the results quite difficult to be understood.

Therefore, I strongly suggest to explain your findings within a clinical point of view (in the discussion).

Author Response

The authors aim to investigate how the hospital exposome, disease progression, and exposome-disease interactions influence physiological functioning, through the untargeted analysis of serum metabolomics of 71 hospitalised moderate and severe COVID-19 patients.

The methods sound correct for the investigation. Manuscript is clear and the language does not need revision.

However, the major limitation for this study lies in the clarity for clinicians. Indeed, physicians not expert in biochemical mechanisms would find the results quite difficult to be understood. Therefore, I strongly suggest to explain your findings within a clinical point of view (in the discussion).

>> Thank you for your valuable insight. We restructured our findings in the discussion to be more in line with a clinical perspective. The updated discussion groups the results by their aspect of potential clinical importance. Specifically, we discuss the results by 1) metabolomic changes related to the hospital exposome, 2) metabolomic changes related to host-microbiome interactions, 3) potential markers of COVID-19 related pathophysiology, and 4) metabolomic changes that deliver potential markers of viral reprogramming of the host metabolism. We hope these changes will make the outcomes and relevance of this manuscript more accessible to clinicians.

Reviewer 3 Report

In this work, Hensen et al study the serum metabolome of a cohort of COVID-19-diseased patients from three Swiss hospitals, using cutting-edge metabolomics platforms and approaches. They identify a plethora of metabolites and compound families, and carry out a deep analysis of the data, trying to find correlations between the changes in metabolites, as well as the use of (some of) them as markers of disease (either progression, severity, or death). From the methodological point of view, the study is well conducted, although the cohort size is small (this is a limitation that the authors recognize at the end of the manuscript), and is excellently written. However, there are some aspects that must be improved to be accepted for publication:

Results:
-Line 119: please revise the numbers of moderate and severe patients from Ticino, as they are not in accordance with the total number of patients from that hospital (i.e. 22) and with shown in Table 1.
-Section 2.3.: the authors detect an important amount of different xenobiotics in the metabolome. They admit that the clinical meaning of those results is not clear. However, how can they be related to disease progression/severity?
-Reference to Figure 3 in the text is missing.
-Sentence in lines 350-352: this is interpretation of results, and should be moved to the Discussion section.

Discussion:
The authors have performed an untargeted metabolomic analysis to detect any possible molecule that can be used as a marker of disease, either for severity, progression, death, etc. They have identified a wide range of metabolites and compound families, among them environmental exposure molecules (xenobiotics). Although of interest for other kind of works in hospitalized patients, it is difficult to understand the possible relationship between this group of molecules and COVID incidence. Nevertheless, the authors recognize that none of the markers showed evidence of a disease-dependent trajectory in blood. Was this a hypothesis before getting the data? About the markers of dietary behaviour, do you think that they can (positive) influence in a better outcome of the disease? I my opinion, the conclusion that these molecules are markers related to hospitalisation cannot be sustained (lines 639-640).

Line 747: “homo sapiens” —> Please write it in italics, and h of homo in capitals.

Please add the description of supplementary materials at page 23, as well as the informed consent statement, data availability statement and conflicts of interest.

Author Response

In this work, Hensen et al study the serum metabolome of a cohort of COVID-19-diseased patients from three Swiss hospitals, using cutting-edge metabolomics platforms and approaches. They identify a plethora of metabolites and compound families, and carry out a deep analysis of the data, trying to find correlations between the changes in metabolites, as well as the use of (some of) them as markers of disease (either progression, severity, or death). From the methodological point of view, the study is well conducted, although the cohort size is small (this is a limitation that the authors recognize at the end of the manuscript), and is excellently written. However, there are some aspects that must be improved to be accepted for publication:

Results:

-Line 119: please revise the numbers of moderate and severe patients from Ticino, as they are not in accordance with the total number of patients from that hospital (i.e. 22) and with shown in Table 1.

>> Thank you for pointing this inconsistency out. We have updated the numbers of moderate and severe patients from Ticino such that they are in line with the correct sample numbers shown in Table 1.

-Section 2.3.: the authors detect an important amount of different xenobiotics in the metabolome. They admit that the clinical meaning of those results is not clear. However, how can they be related to disease progression/severity?

>> The changes in serum xenobiotic concentrations in the first week of hospitalisation suggest changing environmental exposures from the home environment to the hospital environment. This shift towards the hospital exposome can be expected to have systemic effects on human physiology, and may, thus, influence COVID-19 disease progression. However, as no difference could be found in the trajectories of xenobiotic compounds in dependence on either disease severity or mortality (Figure 10, Table S6, Table S7), no link could be found between xenobiotics and disease severity. It is important to note that nowhere in the manuscript, do we state that our results on xenobiotics are linked to disease progression or severity. However, we believe that our results on changing xenobiotic concentrations are still relevant as a step towards a better and more holistic understanding of disease-environment interactions.

-Reference to Figure 3 in the text is missing.

>> The missing reference to Figure 3 has been added.

-Sentence in lines 350-352: this is interpretation of results, and should be moved to the Discussion section.

>> We agree. The sentence has been removed from the results section.

Discussion:
The authors have performed an untargeted metabolomic analysis to detect any possible molecule that can be used as a marker of disease, either for severity, progression, death, etc. They have identified a wide range of metabolites and compound families, among them environmental exposure molecules (xenobiotics). Although of interest for other kind of works in hospitalized patients, it is difficult to understand the possible relationship between this group of molecules and COVID incidence.

>> Thank you for your comment. Investigating the effects of xenobiotics on the incidence of COVID-19 is outside the scope of this study as only COVID-19 cases were included. Our results on xenobiotics suggest changing environmental exposures from the home environment to the hospital environment. The hospital exposome can be expected to have systemic effects on human physiology, which are different from the home environment. Thus, this environmental change may influence COVID-19 disease progression. However, our results could not find evidence for this relationship as no difference could be found in the trajectories of xenobiotic compounds in dependence on either disease severity or mortality (Figure 10, Table S6, Table S7). It is important to note that nowhere in the manuscript, do we state that our results on xenobiotics are linked to disease progression or severity. However, we do believe that our results on changing xenobiotic concentrations are still relevant as part of a step towards a better and more holistic understanding of disease-environment interactions.

Nevertheless, the authors recognize that none of the markers showed evidence of a disease-dependent trajectory in blood. Was this a hypothesis before getting the data? 

>> Thank you for your comment. We would like to note that we found 17 markers of disease-dependent trajectories in the data (Figure 10, Table S7). However, none of these 17 compounds were xenobiotics. In this study, we utilised an untargeted metabolomics approach to obtain a metabolome-wide overview of serum metabolic trajectories in newly hospitalised COVID-19 patients. Before analysing the data, there were no explicit expectations, thus no hypotheses, on which compounds would change over time and which serum compound trajectories would be dependent on disease severity or mortality.

About the markers of dietary behaviour, do you think that they can (positive) influence in a better outcome of the disease? I my opinion, the conclusion that these molecules are markers related to hospitalisation cannot be sustained (lines 639-640).

>> Although several diet-related markers were found to change in newly hospitalised patients, none of these dietary markers were found to be dependent on disease severity or mortality (Figure 10, Table S6, Table S7). Thus, no evidence was found for disease severity or mortality-related serum trajectories of dietary compounds. We would like to note that, similarly to our findings on xenobiotics, these results are still relevant as part of an effort towards holistically characterising all influences on disease progression.

Line 747: “homo sapiens” —> Please write it in italics, and h of homo in capitals.

>> Thank you for this correction. We corrected “homo sapiens” to “Homo sapiens” in the revised manuscript.

Please add the description of supplementary materials at page 23 (1), as well as the informed consent statement (2), data availability statement (3) and conflicts of interest (4).

>> Thank you for these useful comments. The description of figure S1 (1) is updated and completed and the informed consent statement (2), data availability statement (3) and conflicts of interest (4) are added.

Round 2

Reviewer 1 Report

I believe the authors have addressed the questions from the reviewers and the manuscript is ready for publication. 

Reviewer 2 Report

No further comments